# Design of a Cattle-Health-Monitoring System Using Microservices and IoT Devices

**Isak Shabani** [1,*] , **Tonit Biba** [2] **and Betim Çiço** [3]

1 Faculty of Electrical and Computer Engineering, University of Prishtina, 10000 Prishtina, Kosovo
2 Software Development Department, DataProgNet SH.P.K., 70000 Ferizaj, Kosovo; tonit.biba@dataprognet-ks.com
3 Department of Computer Engineering, Epoka University, 1032 Tirana, Albania; bcico@epoka.edu.al
* Correspondence: isak.shabani@uni-pr.edu

**Abstract:** This article proposes a new concept of microservice-based architecture for the future of distributed systems. This architecture is a bridge between Internet-of-Things (IoT) devices and applications that are used to monitor cattle health in real time for the physical and health parameters of cattle, where microservice architecture is introduced that enables this form of monitoring. Within this architecture, machine-learning algorithms were used to predict cattle health and inform farmers about the health of each cattle in real time. Within this architecture, six microservices were proposed that had the tasks of receiving, processing, and sending data upon request. In addition, within the six microservices, a microservice was developed for the prediction of cattle health using algorithms from machine learning using the LightGBM algorithm. Through this algorithm, it is possible to determine the percentage value of the health of each head of cattle in the moment, based on the parameters that are sent from the mobile node. If health problems are identified in the cattle, the architecture notifies the farmer in real time about the problems that the cattle have. Based on the proposed solution, farmers will have 24 h online access to monitor the following parameters for each head of cattle: body temperature, heart rate, humidity, and position.

**Keywords:** agriculture; cattle; cloud; deep learning; docker; IoT; microservices; monolithic; sensor





## 1. Introduction

The influence of technology in recent years and the continued tendency for digitization is being increasingly observed in all areas. Almost every field has digitized parts that enable more efficient and easier management. One of the areas that is being digitized, which is part of the research and subject of the present study, is the field of agriculture, with a particular emphasis on monitoring the health of cattle. In this area, large investments in digitization have been made by state agencies worldwide to ensure higher productivity for farmers. According to the agenda of the World Government summit [1], roughly 800 million people worldwide suffer from hunger and 8% (650 million) of the world's population will still be undernourished by 2030. According to the same report, demand is continuously growing; by 2050, we will need to produce 70% more food.

The field of agriculture, along with industry and services, makes up the main pillars of economic development of every state in the world, and Kosovo is no exception. According to the latest report published by the Ministry of Agriculture, Forestry, and Rural Development, the agriculture sector contributes 8.1% to the gross domestic product (GDP) in Kosovo [2]. According to the same report, 257,733 cattle (calves, appetizers, heifers, bulls, dairy cows, and beef cattle) were reported in Kosovo in 2019, a decrease of 929 heads compared to 2018. In terms of age structure, the cows were younger than 1 year old, 32%, the category aged from 1 to 2 years 10%, and the category older than 2 years old was 58%. Cattle is the most important category included in livestock, and it makes up 47.5% of the total livestock heads [3]. According to [3], in 2020, compared to the previous year, the total

stock of cattle has increased by 1.4%. There was an increase in most categories, except the categories of female calves younger than 1 year old, calves aged 1–2 years, and other cows that marked a decrease in comparison to 2019. In 2020, the total number of cattle was 261,389. According to the same report, by 2020, dairy cows represented 51% of the total number of cattle. The total milk production in 2020 was 281 thousand tons, which is approximately 2% higher than in 2019, because the number of dairy cows was higher. The trade balance remained negative at EUR 29.4 million. Consumption per capita was 170 kg per year, which means that a person consumed approximately 0.5 kg per day, including milk and its by-products.

Many types of diseases occur in cattle, those occurring most often being anthrax (Bacillus anthracis), Aujeszky's disease, bluetongue (BT), bovine coronavirus infection, bovine digital dermatitis, bovine genital campylobacteriosis (Campylobacter fetus subsp. Venerealis), bovine respiratory disease (bovine respiratory syncytial virus), bovine viral diarrhea (BVD), enzootic bovine leukosis (EBL), epizootic hemorrhagic disease, infectious bovine rhinotracheitis (IBR), leptospirosis (Leptospira Hardjo), liver fluke, mycoplasmosis (Mycoplasma bovis), contagious bovine pleuropneumonia (Mycoplasma mycoides subsp. Mycoides SC), neosporosis, paratuberculosis, Q-fever (Coxiella burnetti), salmonellosis, staphylococcal infection (Staphylococcus aureus), streptococcal infection (Streptococcus agalactiae), ringworm (Trichophyton verrucosum), and trichomonosis (Tritrichomonas foetus) [4].

Cattle health is one of the preoccupations of most farmers dealing with growth and cultivation, as the health of cattle has a direct impact on productivity. The physical monitoring of cattle health can be problematic, especially in cases where the numbers are high. To enable the digital monitoring of cattle, a system must be created that initially enables the registration of the physical parameters of the cattle through the deployment of IoT devices to each cattle, the storage and processing of the parameters that are sent, and the final notification of stakeholders if there is a problem with the data that is sent.

The main purpose of this paper is to create an architecture based on the microservices for the specific case of monitoring the health of cattle.

This paper focuses on receiving information from various devices from the Internet of Things, processing them, notifying stakeholders, and providing information to third-party applications that may receive data for their visualization. When we mention third-party applications, we are referring to applications that farmers are currently using in their farms, so the current architecture will allow other applications to obtain information, process it, and display it as needed. To receive information from the IoT devices, microservices will be created that will enable the receipt of data through the REST protocol.

The idea of creating microservices for this case is based on the creation of a distributed system that is scalable at the service level, shares responsibility for different cases, has different programming languages, and the interaction between services [5].

When the information is received from the IoT devices, it should be borne in mind that, to draw the most accurate conclusions, a list of parameters should be sent, from which conclusions will be drawn regarding the current health of the cattle. The list of parameters that must be sent is as follows:

- Body temperature;
- Humidity;
- Heartbeat;
- The position of the cattle.

The position of the livestock is be measured to check if the livestock has been in the same position for a long time. If the cattle do not move for a long time, the farmer will be able to observe in real time why there is no movement, as well as physically observe, first hand, the reasons why the cattle are not moving.

This information should be stored and related to the health of the bovine with the physical activity that the bovine performs.

It is estimated that, by 2020, over 75 million agricultural IoT devices will be in use. The average farm will generate 4.1 million data points daily in 2050, up from 190,000 in 2014 [1].

The architecture created in microservices offers the possibility of accepting the data individually or as a list. This is because the Internet may not always be accessible and, for this reason, the possibility of sending the information as a list of data recorded during the time when there was no Internet should be available. For each parameter, the architecture provides separate microservices.

The microservice-based architecture, in addition to receiving information from IoT devices, includes other important parts, such as:

- Cattle registration;
- Farm registration;
- Registration and login for farmers.

For cattle registration, the system must generate a unique ID that will be used to identify each cow. This unique identification number is placed on the collars of each cow. The sensors are also placed on the bovine collars. The purpose of using bovine collars is to make their placement as easy as possible.

The following sensors were placed on the collars:

- DHT22 AM2302 DHT11/DHT12, Humidity Sensor and Temperature Sensor;
- GY-521 MPU-6050 MPU6050, Module 3 Axis analog gyro sensors;
- SON1205 was used to measure heart rate.

The key objectives of this study are the research and design of a microservice-based architecture for monitoring and predicting the health of cattle.

## 2. Related Works

Microservice architecture is based on the principle of not sharing things [6], which characterizes this architecture for the development and design of software by other architectures. By not sharing things, we mean that every microservice manages its own data, integrity, and consistency. Thus, microservices can manage their own data. This approach is used for the development of distributed systems for which each microservice is independent and executed in its own process [7].

In recent years, many large companies in the world already begun to design their systems as microservices, instead of monolithic architectures, because of the great benefits that this architecture brings. The group of companies that have started to use microservices includes Netflix, Amazon, Guardian, LinkedIn, Spotify, SoundCloud, and Uber [8,9].

Kalia et al. [10] presented a data-centric process for the identification of microservice candidates for migrating legacy software systems into a microservice-based architecture. They used an illustrative example, the World Web dictionary, Elbow, and K-means methods, for the process of microservice identification.

Integrate microservices with IT devices and sensors, perform data operations, integrate several data sources, and seamlessly transfer complex statistical model developments.

In this article, they propose software architecture for livestock monitoring by using the Internet of Things platform, which is based on microservices and the cloud paradigm.

Unold et al. [11] presented a new IoT-based livestock-monitoring system dedicated to the automated measurements of the health state of dairy cow's in a conventional loose-housing cowshed. According to this paper, they developed a system that could monitor the behavior of the dairy cows and allowed them to detect a particular physiological status.

Sharma et al. [12] presented a various wireless sensor network (WSN)-based automatic health-monitoring systems for monitoring various diseases of dairy cattle. According to this paper, they provided a review of the various existing solutions for animal-monitoring systems by using low-power consumption and low-cost sensor nodes.

Suresh et al. [13] presented a system that consisted of data gathering, mobile nodes, and an IoT cloud platform for a cattle-health-monitoring system. According to them,

the system architecture supports the scalability of the data-gathering nodes, which is an essential requirement in terms of real-world deployment.

Saravanan et al. [14] presented a cloud IoT-based livestock management system with three features: animal-healthcare monitoring and recording using IoT sensors via a wearable collar, animal livestock identification using UID for animals, and QR code reading, processing, and display of the details via wireless technology.

From a survey conducted in different countries in 2021, with approximately 950 respondents, it was found that, in large companies, about 85% of them already use microservices to design their systems; however, the number of companies that do not use microservices, but are planning to use them soon, is about 14%, which is a very good figure and shows the increasing interest of companies to use this form of architecture in relation to other architectures. Slightly lower figures, although not by a large number, were found in companies with smaller numbers of employees; however, the percentage of companies that use or are already planning to use microservices is very high [15]. According to the same research, it has been ascertained which programming languages are mostly used for the development of microservices. The results of this paper are shown in Table 1.

**Table 1.** Number of programming languages most used [11].

| Technology | Number of Developers |
|:---:|:---:|
| JavaScript/TypeScript | 437 |
| .Net | 115 |
| PHP | 100 |
| Ruby | 19 |
| Java | 176 |
| Python | 110 |
| Go | 99 |
| Others | 47 |

As can be observed from Table 1 that the microservices are JavaScript and TypeScript, followed by Java and .Net.

Regarding the communication between services, standard communication protocols are mainly used, as in other architectures. Table 2 shows in tabular form the number of communication protocols used by software developers for microservices [16].

**Table 2.** Main communication protocols for microservices [16].

| Communication Protocol | Number of Developers |
|:---:|:---:|
| HTTP | 514 |
| Events | 294 |
| gRPC | 90 |
| Web Sockets | 63 |
| Others | 38 |

The freedom to choose technologies, increased dynamic adaptation, and better handling of complexities are some of the advantages, whereas distribution is a problem. Having many things to distribute means that, for each of them, the executable environment must be downloaded and executed to create the necessary network for the services to communicate with each other and such problems [17]. Currently, the best solutions to this problem are the use of containers (such as Docker) and process automation.

The issues related to the controlled execution for testing and debugging are some of the major disadvantages of microservice architecture. Evaluating and analyzing the

performance of microservices is particularly difficult because it depends on metrics and registers, and is mostly ad hoc, as shown in [16]. Current tools (software) help practitioners to understand the condition and performance of a single microservice, container, or application. However, the interpretation of how all these components work together to identify the main cause of the problems is mainly performed manually. The process of transitioning to a microservice architecture is not straightforward or simple. If a transition occurs, problems with the solution design, support, and operations must be expected [18].

Similar to other architectures, microservices are also characterized by a list of priorities and weaknesses. According to [19], the list of priorities and weaknesses for microservice architecture can be observed in Table 3.

**Table 3.** List of strengths and weaknesses of microservices.

| Priorities | Weaknesses |
| --- | --- |
| Heterogeneity of technologies | Communication between services is complex |
| Resilience | More services, more resources |
| Scalability | Testing |
| Ease of deployment | System deployment challenges |
| Organizational extent | |

However, testing generally distributed and parallel systems is more difficult than testing monolithic systems. Communication between processes plays a more important role in microservice-based applications than in monolithic applications. Monolithic applications can mainly communicate with a small number of external services. For example, a monolithic application might use Google's email service and payment rips. These services have extremely stable APIs. In the case of monolithic communication applications, we can state that the external parts of the application are used [20,21]. Unlike monolithic applications, the communication between processes is a key component of microservices [22]. The microservice-based system is, in principle, a distributed system consisting of microservices, which are essentially APIs that need to communicate with each other to provide the correct service. It is essential to write tests to verify that services interact with each other and with clients. In addition, according to the research conducted with a group of 669 software developers [16], when asked how satisfied developers of microservices are when it comes to maintenance and troubleshooting, they answered with a score of 3.4 out of 5. For the same research question, regarding what the best solutions are for fixing errors in code, we determined that system tracks are the most preferred solution by software developers for identifying errors in code.

Despite the great benefits provided by microservice architecture, security issues remain a hot topic that requires serious treatment in this architecture owing to their distribution nature [23]. In their study, [24] identified 28 good security practices to ensure systems in microservices. These 28 good practices are grouped into six groups: authorization and authentication, tokens and credentials, internal and external microservices, microservices communications, private microservices, and database and environments. The safety standards, which must be implemented according to [25], are:

- OpenID Connect—for the authentication of users;
- OAuth—to limit microservices that can cooperate;
- JWT—to create the identity of each user during communication;
- TLS—to encrypt communication between each service.

As for the comparison with other architectures, comparisons with monolithic architecture [26] and service-oriented architecture (SOA) are most likely to be encountered. According to the results achieved by the work in which monolithic architecture was compared to the microservices-based architecture with communication gates and without a communicative gate, monolithic architecture performed better in the case of the average

response time of the system. According to the same work, it was stated that the results were due to delays that could be caused during communication between the services. The speed of the monolithic application comes from the use of logic for Cache, where several requests that are the same are not processed twice, especially in Get requirements.

In comparison to the SOA architecture, we initially have to say that both architectures are service-based architectures. Generally, these two architectures share many commonalities, as well as a long list of differences. Some microservices critics claim that there is nothing new in microservices compared to SOA [21]. SOA and microservices typically use different technologies. SOA applications typically use "heavy" technologies, such as SOAP and other web-service standards. SOA typically uses ESB, a smart tube that incorporates business logic and message processing to integrate services. Microservices, on the other hand, tend to have much easier communication protocols, such as REST or gRPC [21].

One thing all service-based architectures have in common is that they are generally distributed architectures, meaning that service components are accessed remotely through some sort of remote-access protocol—for example, Representational State Transfer (REST), Simple Object Access Protocol (SOAP), Advanced Message Queuing Protocol (AMQP), Java Message Service (JMS), Microsoft Message Queuing (MSMQ), Remote Method Invocation (RMI), or .NET Remoting [27].

## 3. Model Development and Data Format

In the Smart4ALL project, the digitization of the cattle-health monitoring process was projected through the real-time monitoring of physical parameters and their reporting. Smart4ALL is a four-year innovative action project funded within the projected framework of 2020 [28]. The project envisages communication nodes; cloud infrastructure that will receive, process, and provide information; and a web application that will be used to display the collected data. In the framework of this work, we provide an architecture based on microservices, which is placed in the cloud and provides a means of communication for all entities that need to communicate and exchange information with each other. When we refer to all entities, we mean that we have different types of devices that need to communicate with each other. Mediation in this communication is provided through the microservice architecture presented in this paper.

Through this project, IoT devices enabled the transmission of information in real time regarding the health parameters of cattle. In cases where health parameters cannot be sent, the data were collected on another mobile device, and in the first case with an Internet connection, these parameters were sent for further processing by the microservice architecture. As mentioned above, the IoT system consists of two nodes. The communication nodes were as follows:

- Data-gathering node;
- Mobile node.

The architecture of these two communication nodes and their relationships with the microservices are shown in Figure 1.

As can be observed from Figure 1, the sensors located in the cattle have the task of sending parameters they collect through sensors to the mobile node, while the mobile node must send the parameters for further processing in the cloud. In cloud technology, microservices that provide interfaces for communication with the mobile node through the HTTP protocol are located. The parameters sent by the mobile node are also processed depending on the processing result, and the respective messages are sent to the web application, where the farm can check all cattle in real time.

The web part of the app communicates with microservices to share information that has been sent from the mobile nodes, but also to extract various statistical reports that are necessary for the interested parties.

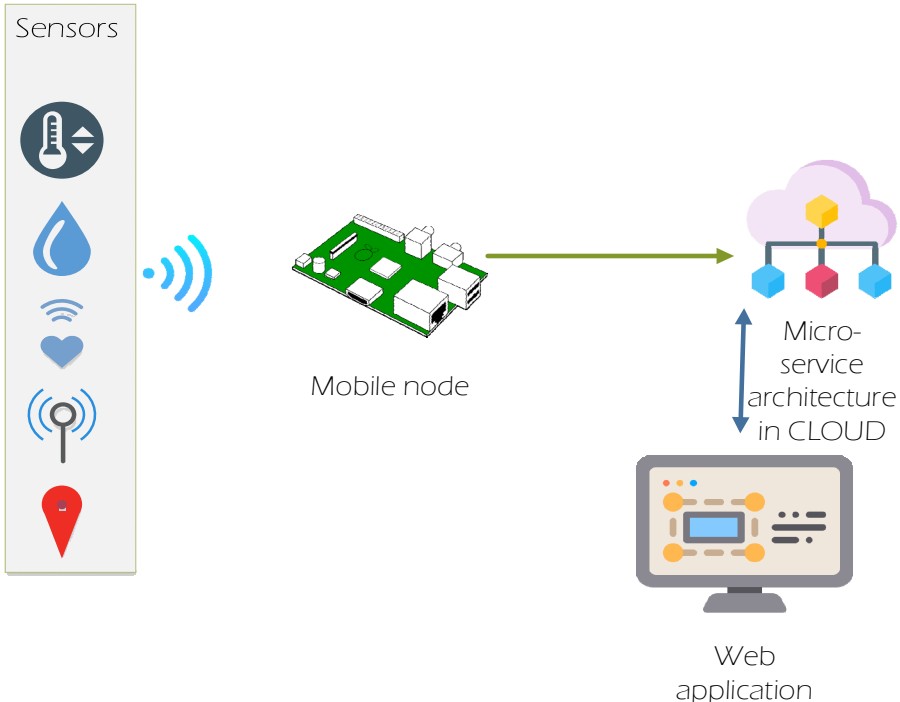

**Figure 1.** System architecture.

The architecture of the microservices was tasked with identifying each device that sent information about specific cattle and the systematization of data sent to the database placed in the cloud technology. Data sent from the mobile node were sent in the JSON format and included information sent by the sensors.

This information was as follows.

- Time when parameters were recorded (year, date, and hour);
- Body temperature (°C);
- Humidity (relative humidity %);
- Heartbeat (number of beats per minute);
- Cattle position (x, y, and z) (meter).

For determining the position of the cattle, we used an accelerometer, which, for this reason, is also the parameter z. The accelerometer returned three values: x, y, and z. The integer values presented in the data format were adjusted and passed to float numbers. The "Date" field was adjusted and we set the name as "Date".

The JSON format of sending this data is shown in Data Format 1.

**Data Format 1.** JSON data acceptance format.

```
{
        CattleID                String nullable: true
        Date                    string ($date-time)
        Pulse                   number ($float)
        Temperature             number ($float)
        X                       number ($float)
        Y                       number ($float)
        Z                       number ($float)
        Humidity                integer ($int32)
}
```

Data sent from mobile nodes to the data-gathering API microservice were received in the format presented in Table 1. As with any RESTful API, this API implemented standard

HTTP protocol methods, such as Get, Post, Put, and Delete. This microservice can accept data individually or as a list of these data. As can be observed in the accepted data structure that the identifying parameter used to uniquely identify each cattle is CattleID, which is in a UUID format [29]. Communication with web applications through relevant microservices is necessary for the administration of farms, farmers, and cattle, as well as for monitoring and issuing various statistical results. Every farmer who has registered cattle must first have a farm. For this purpose, there are two microservices, where one focuses only on farm administration, and the other focuses on administration with farmers. The microservice dedicated to the farm-monitoring section was used to register, update, and delete farms belonging to farmers. The JSON format for recording or updating the data for a given farm is shown in Data Format 2.

**Data Format 2.** JSON format for farm registration.

| | | |
|---|---|---|
| { | | |
| | Name | String nullable: true |
| | Place | integer ($int32) |
| | Address | string nullable: true |
| | FarmerID | number ($float) |
| } | | |

According to Table 2, a farm to be registered must be owned by a farmer, have an address set, and have a name. The other microservice that is dedicated to farmer management is Farmer API, for which microservice aims to register, update, inactivate, and return farmers. The JSON format for updating data for a particular farmer is shown in Data Format 3.

**Data Format 3.** JSON format for registering farmers.

| | | |
|---|---|---|
| { | | |
| | PersonalNumber | string nullable: true |
| | Name | string nullable: true |
| | Surname | string nullable: true |
| | DateofBirth | string ($date-time) |
| | PhoneNumber | string nullable: true |
| | Email | string nullable: true |
| | Password | string nullable: true |
| } | | |

As can be observed from Table 3 to register, a farmer must meet some necessary parameters, such as personal (issued by the government personal identification number for each citizen), name, surname, date of birth, phone number, email, and password.

The idea of using the password is that, when the account is created in the microservice architecture, each farmer can access the part of the data that is for his or her farm. Access to farmers cannot be guaranteed without a password.

Each farmer can own several farms and each farm can contain several cattle. These two microservices are the main ones and constitute the two main pillars that pave the way for the registration of cattle and acceptance of health parameters from the mobile nodes.

Each registered cattle is initially assigned a unique ID, which is then used for acceptance, but is also notified at the level of each cattle.

The structure of the generated ID was in a UUID format. According to this format, each cattle has a code with 36 characters presented in the format 8-4-4-4-12, 32 of which are hexadecimal numbers and 4 hyphens. An example of a unique cattle ID is 12a126ba-0e0d-4476-8d42-4d802231b559. The structure of the cattle identified through UUID and the corresponding QR code are shown in Figure 2.

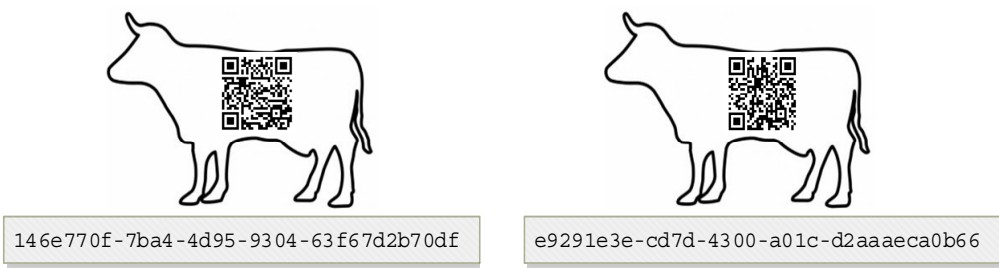

**Figure 2.** Identification of cattle with QR code and UUID.

As shown in Figure 2, each cow has a unique QR code that facilitates the reading of the cattle code and easier access to the data without having to write down the cattle ID.

The QR code is placed in the ear of each bovine as an additional part of the current ethics set by state institutions. Figure 2 is intended to only show that each cattle will have a QR code; the placement is in part of the ear in the form of an earring for each cattle.

The Cattle API microservice is used for cattle registration and the generation of unique IDs. The JSON format for data acceptance is presented in Data Format 4.

**Data Format 4.** JSON format for cattle registration.

```
{
    PersonalNumber          string nullable: true
    FarmID                  integer ($int32)
    Name                    String nullable: true
    Gender                  GenderEnum integer($int32)Enum:[1,2]
    BirthDate               string($date-time)
    BreedID                 integer($int32)
    Weight                  number($double)
}
```

According to the JSON format shown in Table 4, to register cattle that have just been born or purchased from a particular farmer, the farm, cattle name, gender, date of birth, breed, and weight in kilograms must be sent.

This microservice is also responsible for updating the cattle data as well as extracting data for registered cattle. To identify the health problems of cattle, respectively, in the part of forecasting, using machine learning, the system offers the possibility of identifying health problems in a very advanced and very accurate way, regarding the possible complications that may appear in cattle.

## 4. Analysis and Results

Based on the medical knowledge, the normal health parameters of a particular cattle can be identified. Changing the normal parameters automatically indicates that the cattle are becoming sick. It is also well known that, to obtain a more accurate forecast, we must have a relatively large number of monitored parameters.

In this research, the following parameters were analyzed to monitor cattle health: body temperature, humidity, heartbeat, and position. Based on the results of the scientific studies for the field of cattle medicine, parameters, such as temperature and heartbeat, have a normal range of values; according to [30], these values, are presented in Table 4.

**Table 4.** Normal cattle parameters.

| Vital Parameters | Normal Value |
|---|---|
| Temperature | 37.78–39.17 |
| Heartbeat | 100–140 beating per minute |
| Respiratory rate | 30–60 breathing per minute |

Based on the normal parameters shown in Table 4, the results can be obtained that indicate whether the cattle are within the normal parameters.

However, in the future, these parameters can be expanded with additional parameters, which will enable even more accurate predictions than those calculated with these parameters. A cattle that was monitored. Cattle measurements were performed hourly.

The randomly generated dataset for cattle-health findings is shown in Table 5. The number of rows generated for the model development was 10,000. A total of 60% of the dataset was used for model training, 20% for test set, and 20% for validation set.

**Table 5.** Data collection for cattle parameters.

| Temperature (°C) | Pulse (bpm) | Humidity | X | Y | Z | Date | Time | Health |
|---|---|---|---|---|---|---|---|---|
| 38.5 | 111 | 6 | 933 | 1060 | 892 | 2021-10-01 | 10:30:00 | 84% |
| 38.5 | 140 | 97 | 629 | 968 | 641 | 2021-10-01 | 23:39:38 | 83% |
| 38.4 | 80 | 68 | 969 | 906 | 96 | 2021-10-02 | 10:05:35 | 80% |
| 38.05 | 107 | 7 | 272 | 18 | 676 | 2021-10-03 | 06:32:27 | 92% |
| 39 | 125 | 39 | 1073 | 1094 | 576 | 2021-10-03 | 14:28:06 | 86% |
| 38.05 | 83 | 88 | 426 | 475 | 988 | 2021-10-04 | 12:04:38 | 88% |
| 37.9 | 151 | 92 | 844 | 203 | 880 | 2021-10-04 | 15:16:48 | 88% |
| 38 | 150 | 92 | 398 | 976 | 294 | 2021-10-05 | 06:56:34 | 87% |
| 37.9 | 148 | 14 | 454 | 1092 | 629 | 2021-10-05 | 20:58:40 | 96% |
| 38.01 | 158 | 77 | 839 | 816 | 498 | 2021-10-06 | 09:20:29 | 84% |
| 39 | 117 | 29 | 1080 | 915 | 720 | 2021-10-07 | 08:37:37 | 79% |
| 38.89 | 117 | 2 | 489 | 42 | 261 | 2021-10-07 | 11:43:43 | 78% |
| 38.64 | 131 | 91 | 93 | 114 | 1075 | 2021-10-08 | 10:30:53 | 91% |
| 38.5 | 155 | 45 | 155 | 665 | 926 | 2021-10-08 | 15:38:51 | 88% |
| 38.5 | 134 | 75 | 618 | 339 | 348 | 2021-10-09 | 05:24:10 | 87% |
| 39 | 134 | 98 | 701 | 338 | 781 | 2021-10-09 | 12:11:19 | 78% |
| 38 | 113 | 67 | 149 | 339 | 1085 | 2021-10-09 | 20:40:17 | 92% |

The health column shows the percentage of bovine health from 0 to 100%, so that 100% is taken as the value where the bovine has all the parameters in order. This column was calculated by considering the normal values of bovine health according to Table 4, putting into function the parameters of air humidity and bovine position. As can be observed from Table 5, all data, except for the last column, are generated by mobile nodes and sent to the microservice architecture for further processing. Based on the parameters sent and using the preliminary data, the value of the last column can be predicted in real time for each cow. The methods and algorithms from a variety of machine-learning methods can be used for this purpose.

In this case, a machine-learning algorithm was used. This algorithm was a light-gradient-boosting decision tree (LightGBM). Before choosing this algorithm, other algorithms in this field were tested, and the results were derived, as presented in Table 6.

The generated dataset was placed in the open-source ML.NET framework. This framework listed the seven algorithms that were the most suitable for creating a model for predicting bovine health. Of these seven algorithms, we placed three of them in the short list of algorithms that should be used to predict bovine health. From the results obtained from this framework and based on the four key parameters, it was estimated that the LightGBM algorithm was the most suitable for this purpose. The following are the

parameters with which the comparisons were made: R-squared, absolute loss, squared loss, and root-mean-square error.

**Table 6.** Normal cattle parameters.

| Algorithm | R-Squared | Absolute Loss | Squared Loss | RMS Loss |
|---|---|---|---|---|
| LightGbmRegression | −0.0055 | 24.56 | 806.42 | 28.38 |
| SdcaRegression | −0.1410 | 25.73 | 912.88 | 30.18 |
| FastForestRegression | −0.0386 | 24.73 | 832.92 | 28.83 |

The ML.NET library created three models for the generated dataset, as presented in Table 6. These models were selected from a library.

These results were generated by the ML.Net package for the cattle dataset, and the parameters are shown in Table 5. The parameters presented in Table 6 have the following meanings:

- R-squared—represents the power of forecasting the model as a value in the range [−∞, 1.00]. The value of 1.00 means that the model had a perfect fit. The 0.00 result means that the model guessed the expected value. This coefficient measured the closeness of the test-data values to the predicted values [31]. Formula (1) for calculating this coefficient is:

$$KP = 1 - \frac{SS_{res}}{SS_{tot}} \tag{1}$$

where: $SS_{res}$—Is the sum of squares due to regression. $SS_{tot}$—It is the total number of squares.

In the concrete case, the value for the LightGbmRegression algorithm was calculated (−0.0055), which was a higher value compared to the other two algorithms, and the algorithm performed better than the other two algorithms.

- Absolute loss—measures the closeness of the forecasts to the actual results. This coefficient represents the average of all model errors, where the model error was the absolute distance between the predicted health value (in this case) and the correct value. The closer the calculated value was to zero (0.00), the better the quality [31]. Formula (2) for calculating the average absolute error is [28]:

$$MGA = \frac{1}{N} \sum_{i=1}^{N} |Y_i - \hat{Y}_i| \tag{2}$$

According to the formula above, it can be concluded that the mean of the absolute error is the sum of the absolute values of the difference between the calculated and predicted values. In this case, the calculation with the LightGbmRegression algorithm was 24.56, which means that the deviation from the real estimate was ±24.56. This value means that the predicted distance of the correct value was 24.56, regardless of whether it was in the positive or negative part of the line. Because the value of 24.56 is closer to zero (0.00), this indicates that the LightGbmRegression algorithm has a lower mean absolute error, and is therefore chosen as the most suitable algorithm in this respect.

- Squared loss; in other words, the mean deviation squared—indicates how close a regression line is to a set of values by taking the distances from the points on the regression line. The square adds more weight to the large differences [27]. As with the mean absolute error, the values that are closer to zero indicate a higher quality in the forecast. Formula (3) for calculating the average error in squares is [32].

$$GMK = \frac{1}{N} \sum_{i=1}^{N} (Y_i - \hat{Y}_i)^2 \tag{3}$$

For the present case, the LightGbmRegression algorithm has a value of 806.42, considering the definition that the values that are closer to zero indicate a higher quality. Based on the formula shown above, it can be concluded that the LightGbmRegression algorithm is more suitable for this aspect.

- Root-mean-Square error—represents the difference between the values predicted by a model and the values observed by the environment being modeled. The mean-root-square error is mainly used for predictions and regression analysis to verify experimental results. Values that are closer to zero (0.00) indicate a higher quality in the forecast [31]. The Formula (4) for calculating the mean deviation to the square root is [32]:

$$DMRK = \sqrt{\frac{1}{N} \sum_{i=1}^{N} (Y_i - \hat{Y}_i)^2} \tag{4}$$

For the present case, the LightGbmRegression algorithm had a value of 28.38, considering the definition that the values that are closer to zero indicate a higher quality. Based on the formula shown above, it can be concluded that the LightGbmRegression algorithm is more suitable in this respect.

Based on the results shown in Table 6, it can be concluded that the most suitable algorithm for predicting cattle health is LightGbmRegression.

After modeling, training, and evaluation, the system was ready to predict cattle health based on the parameters sent by the mobile node. For this purpose, a microservice was created that offered the possibility of sending the parameters introduced by the mobile node and returned the forecast result to a percentage of 0–100. The JSON format for receiving data for this microservice is shown in Data Format 5.

**Data Format 5.** JSON format for data forecasting with LightGBM.

```
{
        Temperature                     number($float)
        Pulse                           number($float)
        Humidity                        number($float)
        X                               number($float)
        Y                               number($float)
        Z                               number($float)
        Date                            string nullable: true
        Time                            string nullable: true
}
```

The moment this microservice returns as a result a value that is low, it is automatically responsible for sending an email to the farmer to notify him/her that something is wrong with the health parameters of the individual cattle.

The proposed architecture for monitoring the health of cattle is shown in Figure 3. The architecture shown in Figure 3 was implemented using the C # programming language with the Asp.NET Core Web API framework. The architecture shown in Figure 3 combines the microservices shown above by adding two microservices that were not treated above, the microservices to guarantee access control and the communication gateway used for intermediation between IoT devices and the web application, with the part of the microservices that are implemented. This architecture enables farmers through web or mobile applications to access the system for monitoring the health of cattle and to register their farms and cattle.

The unstructured data were obtained from the sensors and then they were used in the application of cattle between microservices and the IoT after the serialization of the data was made readable by farmers; in this way, the data were retrieved.

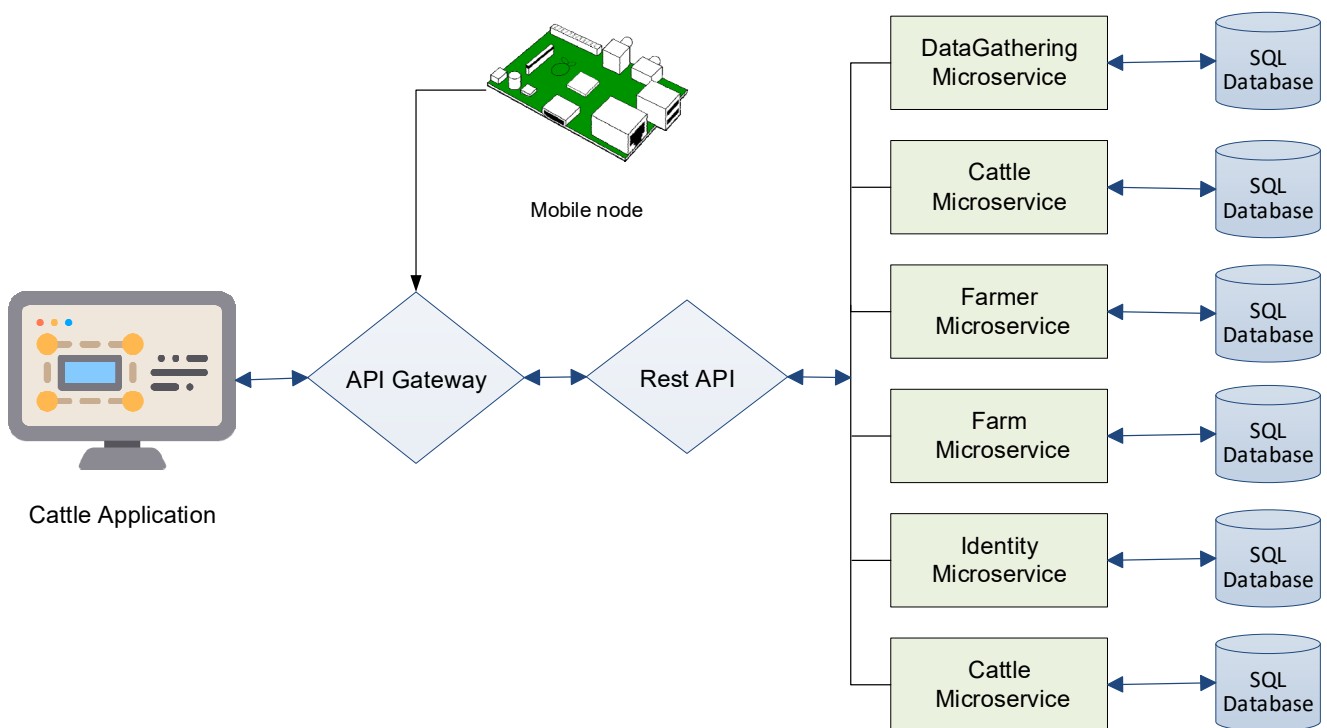

**Figure 3.** Architecture for cattle-health monitoring.

Health values for cattle were obtained after many measurements were conducted. The root-mean-square error represents the difference between the values predicted by a model and the values observed by the environment being modeled. The root-mean-square error is mainly used for predictions and re-grissino analysis to verify the experimental results. During the results-gain experiments, two farms with 15 and 20 cattle were used.

As can be observed from Figure 3, the API gateway handles requests from others in the same way as a proxy server would in the reverse direction. In this scenario, instead of exposing all the links to public access, we can hide them behind the API gateway. In addition, in this case, the API gateway implements a security standard that prevents others from misusing microservices, balancing requests, aggregate requests, quality of service, logging and tracing, authentication, and authorization. In Figure 3, Identity API is simply an IdentityServer4 responsible for handling authentication and authorization for all microservices.

The API gateway was developed as a light-web application that receives requests from end-users through the Internet and consumes the private services offered by the six microservices through the Rest API. The message interchange protocol connects the browsers to the gateway and the gateway to each microservice in a JSON format.

The cattle-health-monitoring system was built using an MVC pattern design, API gateway, Rest Web API, and microservices for database communication. The gateway did not store any information. The microservice architecture was deployed on the cattle-monitoring cloud platform using the deployment illustrated in Figure 3.

We based this research on the real, tested documented results of the performance of our own cloud-based software application based on microservices [33], proposed a model-driven approach to re-engineering legacy systems into cloud-oriented services. The efficacy of this approach was demonstrated by modeling real-world re-engineering scenarios and obtaining user feedback. This approach provided the basis for the consistent creation, representation, and maintenance of re-engineering services within cloud computing [34].

Mateo-Fornés et al. [35] presented a platform to integrate new sensor devices, perform data operations, integrate several data sources, seamlessly transfer complex statistical model developments, and provide a user-friendly graphical interface. In this paper, they

proposed software architecture for livestock monitoring by using the Internet of Things platform, which is based on microservices and the cloud paradigm.

The parameters were monitored using sensors placed on the bovine collar. These sensors were obliged to send the parameters for further processing at intervals of five seconds. After obtaining these parameters, we continued to predict the health of the cattle and identify the potential disorders that could occur in cattle.

While using cloud technology, it is possible to extend it to almost every geographical region by simply having access to the Internet. By using cloud technology, it is possible to place the same instance in different geographical regions. Each microservice is placed in a Docker container, where the container orchestration is performed by the Kubernetes. Given the fact that the cattle-health-monitoring system must have an extremely high degree of availability, mobile nodes that provide data from sensors placed on the cattle will not always have Internet access to send such data.

Two methods were used to achieve this goal in real time. The first was using web sockets, respectively, with the SignalR library. In real time, when the microservice for the prediction of cattle health found that the evaluation of the accepted parameters did not provide a satisfactory result, it had the task of establishing a connection with web sockets and sending a message to the troubled cow. Every message sent through web sockets was reflected in real time, on the front page of the web application. The second method was by sending an email to the farmer; this path was used, especially when the connection for communication with the web sockets could not be made.

Microservices were utilized to create a communication bridge between the mobile nodes responsible for receiving information and the part of the web application used by the farmers and other stakeholders. The created microservices were responsible for accepting the health parameters, uniquely identifying each one, systematizing the information stored in the databases, and predicting the health to inform the farmer about the cases when, based on the forecast with algorithms from machine learning, they were found to have health problems.

Therefore, creating a satisfactory level of system availability was essential, which was achieved by creating copies of microservices that were critical to providing services. This was achieved by using new technology and Kubernetes, which enabled the use of hardware resources on a large scale, creating duplicates of critical services and expanding general resources, if necessary.

## 5. Conclusions

In this study, we proposed and developed a microservice-based architecture that could serve as a link between IoT devices and various applications that consume the data processed by this architecture. In principle, the proposed architecture was used to monitor the health of cattle, but such architecture can be freely used for areas, such as the respective adaptations.

In this architecture, it is assumed that each cow has a sensor placed on its collar, where each of these sensors collects data on the physical and health parameters of the cow. Body temperature, humidity, heart rate, and cattle position were included in the list of these parameters. The list of parameters for cattle collected from the mobile nodes was readily accepted by the architecture developed in the microservices. In the future, this list of parameters can be expanded using additional parameters. Within this architecture, six microservices were proposed that had the tasks of receiving, processing, and sending data upon request.

In addition, within the six microservices, a microservice was developed for the prediction of cattle health using algorithms from machine learning using the LightGBM algorithm. Through this algorithm, it was possible to determine the percentage value of the health of each cattle at the moment the parameters were sent from the mobile node. If health problems were identified in the cattle, the architecture notified the farmer in real time about the problems that the cattle had.

From the measurements that were made within 24 h, the values in the range limits for each cattle parameter were reached: minimum body temperature 37.9 °C and maximum 39 °C, heart rate from 80 bpm to 158 bpm, humidity from 2% to 98%, and general cattle health from 78% to 96%.

For the measurements obtained from the farm with 15 cattle, the average of the values for each parameter were body temperature 38.20 °C, heart rate 120 bpm, humidity 76%, and general cattle health from 88%.

For the measurements obtained from the farm with 20 cattle, the average of the values for each parameter were body temperature 38.50 °C, heart rate 128 bpm, humidity 82%, and general cattle health from 94%.

For the measurements obtained from the two farms with 15 and 20 cattle, the average values for the cattle in the two farms within 24 h were body temperature 38.40 °C, heart rate 125 bpm, humidity 80%, and general cattle health 92%.

This article contributes to the creation of microservice-based systems that are robust, scalable, reliable, fault tolerant, and enforceable.

This article referred to the Smart4All project where we presented an innovative approach to the development of microservices for the bovine-health-monitoring system. Such a system helps farmers greatly, and assists state bodies in identifying the number of cattle, their type, and sex. This developed architecture enables farmers to access existing applications and access information.

In contrast to the existing work in this area, we introduced a broader approach in terms of service delivery and accessibility for a larger number of users who need to receive real-time information from cattle monitoring in real time.

In this article, with the developed system, we were not limited to the use of a specific type of sensor, but could use any type of sensor from any manufacturer. We were not limited to one application, but provided the interface by which any existing application could be accessed on the sole condition that it had the relevant authorizations to access the data to which it belonged.

**Author Contributions:** Conceptualization, I.S. and T.B.; methodology, I.S. and B.Ç.; software, I.S. and T.B.; validation, T.B.; formal analysis, I.S. and T.B.; resources, I.S. and T.B.; data curation, T.B.; writing—original draft preparation, I.S. and T.B.; writing—review and editing, I.S. and B.Ç.; visualization, T.B.; supervision, I.S. and B.Ç.; project administration, I.S.; funding acquisition, I.S. and B.Ç. All authors have read and agreed to the published version of the manuscript.

**Funding:** This research was funded by the Horizon 2020 Framework Program for Smart Anything Everywhere, grant number No. 872614, which supported this research and provided all the relevant information, work on task 5.6. Design Services Access and Support. Link for project https://smart4all-project.eu (accessed on 1 April 2022).

**Institutional Review Board Statement:** Not applicable.

**Informed Consent Statement:** Not applicable.

**Data Availability Statement:** Data are contained within the article.

**Acknowledgments:** The author would like to thank all universities, SME, and NGO from twenty-five partners from different countries in Europe, which are part of the consortium of Smart4All research and the innovation action project, and we would also like to thank all the participants who contributed their knowledge and insights to this academic and industrial research.

**Conflicts of Interest:** The authors declare no conflict of interest.

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
