# Peer review of "Design of a Cattle-Health-Monitoring System Using Microservices and IoT Devices"

_computers, doi:10.3390/computers11050079_

Round 1
Reviewer 1 Report
1.The article addresses a concept of microservice-based architecture for the future of distributed systems including its application for cattle health monitoring system.
2.The considered architecture involves and provides the following:
(i) a bridge between Internet of Things (IoT) devices and applications that are used to monitor cattle health in real-time for the physical and health parameters of each cattle;
(ii) machine learning algorithms which are used to predict cattle health and inform farmers about the health of each cattle in real-time;
(iii) six microservices for the tasks of receiving, processing, and sending data upon request;
(iv) an additional microservice for the prediction of cattle health using algorithms from machine learning using the LightGBM algorithm.
3.The article material can be used for various other applications.
4.The presentation of the article material is very good (including description of the proposed architecture, methods, application, illustrations).
5.The article can be accepted (as it is).
Author Response
Dear Reviewer 1,
We would like to thank you very mach for the useful comments to improve the our paper with manuscript ID: computers-1720433, entitled “Design of cattle health monitoring system using microservices and IoT devices” and for your valuable comments and suggestions, which we have carefully analyzed and addressed accordingly in the revised version after reviews of our manuscript.
Responses for Reviewer 1
We made all corrections and tried to improve all the addressed points on this revision manuscript (after reviews) as follows:
Point 1: The article addresses a concept of microservice-based architecture for the future of distributed systems including its application for cattle health monitoring system?
Response 1:
We agree with this comments and suggestions.
Point 2: The considered architecture involves and provides the following:
(i) a bridge between Internet of Things (IoT) devices and applications that are used to monitor cattle health in real-time for the physical and health parameters of each cattle;
(ii) machine learning algorithms which are used to predict cattle health and inform farmers about the health of each cattle in real-time;
(iii) six microservices for the tasks of receiving, processing, and sending data upon request;
(iv) an additional microservice for the prediction of cattle health using algorithms from machine learning using the LightGBM algorithm.
Response 2:
We agree with this comments and suggestions. All these suggestions and comments are included and impenetrated in the article. Thank you for thise suggestions.
Point 3: L74: The article material can be used for various other applications.
Response 3:
We agree with this comments and suggestions.
Point 4: The presentation of the article material is very good (including description of the proposed architecture, methods, application, illustrations).
Response 4:
We agree with this comments and suggestions.
Point 5: The article can be accepted (as it is).
Response 5:
We agree with this comments and suggestions.

Reviewer 2 Report
Monitoring the current health status of cattle is a very important task in agriculture. The quality of dairy and meat products depends on this. Uncontrolled outbreaks of infectious diseases in large herds of cattle lead to large financial losses. That is why the results of this study are very important and relevant. The article is well written and contains interesting things. However, I have some comments.
1) More discussions and literature review should be added to the introduction.
2) The authors should explain why they have chosen the parameters (body temperature, humidity, heartbeat, and the position of the cattle) for Cattle health monitoring.
3) The whole paper is too verbose. Very negligible technical things are presented and nothing new.
4) This paper has no aim. The aim should be outlined in the introduction and it should be mentioned in the conclusion whether the aim was reached.
5) The paper present no scientific novelty, most of the results derived by authors are well known in theory. Problem statement should be more justified.
6) Abstract and conclusion is not up to the mark. No proper technical approach. No numerical results are displayed in the Abstract and Conclusion.
7) There are some typos. In line 378 there is an unfinished sentence: «The column health…».
In general, the article is good. Minor correction is needed. The content of the article is consistent with the scientific area of the Computers journal.
Author Response
Dear Reviewer 2,
We would like to thank you very much for the useful comments to improve the our paper with manuscript ID: computers-1720433, entitled “Design of cattle health monitoring system using microservices and IoT devices” and for your valuable comments and suggestions, which we have carefully analyzed and addressed according in the revised version after reviews of our manuscript.
Responses for Reviewer 2
We made all corrections and tried to improve all the addressed points on this revision manuscript (after reviews) as follows:
Point 1: More discussions and literature review should be added to the introduction.
Response 1:
In the literature review, four related papers have been added to address the developments that have been carried out in the field of online cattle health monitoring.
Point 2: The authors should explain why they have chosen the parameters (body temperature, humidity, heartbeat, and the position of the cattle) for Cattle health monitoring.
Response 2:
In the papers: (Cattle Health and Environment Monitoring System D. Aswini, S. Santhya, T. Shri Nandheni, N. Sukirthini) and (Cattle Health Monitoring System Using Arduino and IoT J.Tamilselvan M.E, M.Naveenkumar, K. Periyapandi, B. Premkumar) are used to treat various diseases that can affect the health of cattle as well as their symptoms. These diseases can cause changes in the parameters of the cattle body, which can be identified, and sensors can effectively detect these changes. The table below shows the connection between the sensors and body parameters according to a previous study (Akhila Suresh and T V Sarath 2019 IOP Conf. Ser .: Mater. Sci. Eng. 561 012106).
Parameters to be measured |
Temperature |
Fall detection |
Heart rate |
Humidity |
Sensors for measuring the parameters |
NTC Thermistor |
Accelerometer |
Pulse sensor |
DHT11 |
Diseases mapped to the change in the body parameter |
Fever Ovarian Cyst |
Ovarian Cyst Milk Fever |
Stress Anxiety |
Stress level |
Point 3: The whole paper is too verbose. Very negligible technical things are presented and nothing new.
Response 3:
We have treated the paper theoretically, and we believe that there are findings in this paper. One of the findings is that we have used modern (current) microservices instead of traditional Web services and Web APIs. In this paper we do not intend to explain microservices only theoretically, the purpose of this paper is to develop the architecture of microservices for a specific case, such as the case of monitoring the health of cattle that you can find in this paper.
Point 4: This paper has no aim. The aim should be outlined in the introduction and it should be mentioned in the conclusion whether the aim was reached.
Response 4:
We have added the purpose of the paper as a paragraph in the Introduction. Thank you for this suggestion.
Point 5: The paper present no scientific novelty, most of the results derived by authors are well known in theory. Problem statement should be more justified.
Response 5:
This paper presents a scientific approach to the process of designing an architecture based on microservices for monitoring cattle health. The aim of this study was to create an architecture based on microservices. Unlike other papers that we have referred to in this paper, our work offers a different approach to the cattle health monitoring system. In this study, the number of sensors or their types was not important. It is important that the developed architecture provide an opportunity to receive such information.
Point 6: Abstract and conclusion is not up to the mark. No proper technical approach. No numerical results are displayed in the Abstract and Conclusion.
Response 6:
The result of this paper is the architecture developed based on microservices for the process of monitoring the health of cattle. The whole work focuses on the development of this architecture, which is able to receive information from sensors of different types, to register farms, farmers, and cattle, and to predict the health of cattle.
Point 7: There are some typos. In line 378 there is an unfinished sentence: «The column health…».
Response 7:
We removed this sentence from the revised manuscript. Thank you for this suggestion.

Reviewer 3 Report
The work is devoted to the development of the system based on microservices architecture and IoT technology for the problem of cattle health monitoring. Being technically sound the manuscript has a lot of shortcomings that should be addressed to be accepted for publication.
Comments:
- Microservices are the keystone of the work. Please list and describe the microservices. You mention microservices in the work a few times but you do not call their names. Fig. 3 does not show microservices in a clear way. "Farm API", "Cattle API", "Farmer API" are not microservices. Microservices should describe "activities" or "process".
- line 369-375. The meaning of the "health" column should be described clearly, with the help of mathematical expression.
- What is the practical sense of the usage of the "health" indicator in such a form? (the last column of Table 5) Have the indices from Table 4 be scaled and normalized in order to calculate "health"?
- Expressions (1)-(4) (especially 4) are well known and can be omitted.
- line 362-365. The training of the model is unclear. What resampling strategy was used? Is it a 5-fold cross-validation? What do you mean "Sixty% of dataset 363 was used for model training, 20% for model development"? What is the difference between "model training" and "model development"?
- Please describe the machine learning problem properly. What is the target attribute? Health? As far as I see, you solve the regression problem with the help of the LightGBM decision tree. Please show the decision tree in a plot. It is unclear the usage of the decision tree for solving the regression problem. Please describe it.
Author Response
Dear Reviewer 3,
We would like to thank you very much for the useful comments to improve the our paper with manuscript ID: computers-1720433, entitled “Design of cattle health monitoring system using microservices and IoT devices” and for your valuable comments and suggestions, which we have carefully analyzed and addressed accordingly in the revised version after reviews of our manuscript.
Responses for Reviewer 3
We made all corrections and tried to improve all the addressed points on this revision manuscript (after reviews) as follows:
Point 1: Microservices are the keystone of the work. Please list and describe the microservices. You mention microservices in the work a few times but you do not call their names. Fig. 3 does not show microservices in a clear way. "Farm API", "Cattle API", "Farmer API" are not microservices. Microservices should describe "activities" or "process".
Response 1:
We have redesigned Fig. 3; now, microservics can be seen in a cleaner way.
FarmAPI, CattleAPI, and FarmerAPI are microservices developed for specific purposes. FarmAPI has a single purpose, registration, change, and passivation of farms. As shown in the literature on microservices, each microservice must perform a single job in the proper form. By following this principle, we can divide the part of the system that has a low degree of cohesion into small services that work independently of each other and are executed in their own process.
The API gateway was developed as a light web application that receives requests from end-users through the Internet and consumes the private services offered by the six microservices through the Rest API. The message interchange protocol connects the browsers to the gateway and the gateway to each microservice in JSON format.
A cattle monitoring system was built using MVC pattern design, API Gateway, Rest Web API, and microservices for database communication. The gateway did not store any information. The microservice architecture was deployed on the cattle monitoring cloud platform using the deployment illustrated in Figure 3.
Point 2: line 369-375. The meaning of the "health" column should be described clearly, with the help of mathematical expression.
Response 2:
Yes, the health column that we needed to create the model from machine learning is related to table 4. Table 4 shows the range of normal values that the cattle must be considered healthy. Deviations from these values indicate poor cattle health. We have tried based on the normal values from table 4 to show the health of cattle, which range from 0% to 100%. A percentage calculation is not necessary at this stage. By using the proposed architecture in microservices under real conditions, this model can be further improved.
Point 3: What is the practical sense of the usage of the "health" indicator in such a form? (the last column of Table 5) Have the indices from Table 4 be scaled and normalized in order to calculate "health"?
Response 3:
The health column is calculated based on the normal values from this table. The main purpose of this study was to develop an architecture based on microservices.
Point 4: Expressions (1)-(4) (especially 4) are well known and can be omitted.
Response 4:
Because we used these formulas for the testing part of the machine learning algorithms, we have considered it reasonable to mention them in this paper.
Point 5: line 362-365. The training of the model is unclear. What resampling strategy was used? Is it a 5-fold cross-validation? What do you mean "Sixty% of dataset 363 was used for model training, 20% for model development"? What is the difference between "model training" and "model development"?
Response 5:
A ready-made library from Microsoft ML.Net was used to train the model, and the results are shown in Table 6. Using this library with the dataset shown in table 5, the algorithms shown in Table 6 we tested. Sixty% refers to 60% of the dataset. In the manuscript, we have adjusted the 60% parts and the partition part of the dataset.
Point 6: Please describe the machine learning problem properly. What is the target attribute? Health? As far as I see, you solve the regression problem with the help of the LightGBM decision tree. Please show the decision tree in a plot. It is unclear the usage of the decision tree for solving the regression problem. Please describe it.
Response 6:
The problem addressed by machine learning are related to the prediction cattle health. All columns except the health column come from the sensors, which are then transmitted by the mobile node. Based on Table 4 the normal values of the parameters obtained from the sensors can be ascertained. In the papers: (Cattle Health and Environment Monitoring System D. Aswini, S. Santhya, T. Shri Nandheni, N. Sukirthini) and (Cattle Health Monitoring System Using Arduino and IOT J. Tamilselvan M.E, M. Naveenkumar, K. Periyapandi, B. Premkumar) are treated with various diseases that can affect the health of cattle as well as their symptoms.
These diseases can cause changes in the parameters of the cattle body, which can be identified, and sensors can effectively detect these changes. The table below shows the connection between the sensors and body parameters according to a previous study (Akhila Suresh and T V Sarath 2019 IOP Conf. Ser .: Mater. Sci. Eng. 561 012106).
Parameters to be measured |
Temperature |
Fall detection |
Heart rate |
Humidity |
Sensors for measuring the parameters |
NTC Thermistor |
Accelerometer |
Pulse sensor |
DHT11 |
Diseases mapped to the change in the body parameter |
Fever Ovarian Cyst |
Ovarian Cyst Milk Fever |
Stress Anxiety |
Stress level |
In the problem shown in the paper where Machine Learning is used the target attribute that is used is 'Health' all other fields come from the sensors.
The decision tree is not presented in the paper because the ready-made library from Microsoft ML.NET is used, which, in accordance with the provided dataset, generates the results and creates the model.

Round 2
Reviewer 3 Report
All my comments have been addressed